# Hyperbolic curvature as an inductive bias for latent space flow matching

**Federica Valeau**
University of Amsterdam
federica.valeau@student.uva.nl

**Maria Esteban Casadevall**
AMLab, University of Amsterdam
m.estebancasadevall@uva.nl

**Erik J. Bekkers**
AMLab, University of Amsterdam
e.j.bekkers@uva.nl

## Abstract

Learning image representations that respect the intrinsic geometry of data is crucial for capturing hierarchical semantic structure, yet generative transport is typically performed in Euclidean spaces where this structure is not preserved. In this work, we propose a geometry-aware generative framework that combines hyperbolic representation learning with Riemannian Flow Matching to perform generative transport directly in hyperbolic latent space. Instead of learning generative dynamics in pixel space or Euclidean latents, we transport samples directly on the manifold produced by a pretrained hyperbolic autoencoder, preserving geometric organization and yielding more stable samples and improved FID score, compared to Euclidean latent transport. We further investigate curvature as a controllable geometric inductive bias and observe a trade-off between generation realism and diversity, where moderate curvature yields more coherent samples, and larger curvature allows visual variation at the cost of stability, highlighting how latent geometry shapes generative transport.

## 1 Introduction

Natural images exhibit hierarchical structure across multiple semantic levels, with each image naturally decomposing into attributes of different levels of granularity. Hyperbolic spaces are a natural geometric framework for encoding such hierarchies due to their exponential volume growth. Recent works have shown that learning image representations in hyperbolic space leads to latent organizations that better reflect semantic hierarchies, compared to Euclidean embeddings (Pal et al., 2025). In particular, Hyperbolic Autoencoders have demonstrated strong performance in few-shot image generation (Li et al., 2023; 2025), zero-shot recognition (Liu et al., 2020; Wang et al., 2025), and multimodal tasks (Desai et al., 2023; Pal et al., 2025) by learning geometry-aware latent representations that better reflect semantic hierarchies than Euclidean embeddings.

At the same time, image generation has seen rapid progress in recent years, with state-of-the-art models aiming to achieve both high image quality and fine-grained controllability. In particular, flow matching (Lipman et al., 2023) provides a deterministic alternative to diffusion-based models by learning continuous vector fields that transport a simple base distribution (usually a Gaussian) to the data distribution. Recent works have extended flow matching in non-Euclidean spaces (Chen & Lipman, 2024; Zaghen et al., 2026; Kapusniak et al., 2024), including hyperbolic manifolds. However, to the best of our knowledge, these approaches have not been tested yet in Riemannian latent spaces generated from non-Euclidean autoencoders. Moreover, performing flow-based generation directly in pixel space is computationally expensive due to its considerably high dimensionality (Dao et al., 2023). Latent-space generative modeling enables efficient sampling while retaining semantic structure. More details on related works can be found in Appendix A.1. To address these gaps, in this paper we provide the following contributions:

(i) We formulate generative modeling as *Riemannian flow matching* on the Poincaré ball latent space learned by a hyperbolic autoencoder. Our approach ensures that the transport dynamics evolve intrinsically on the manifold rather than in an ambient Euclidean space. Sampling from the learned flow produces latent representations that respect the underlying hyperbolic geometry and can be decoded into images, enabling image generation without retraining the representation model.

(ii) By adjusting the magnitude of the curvature, we obtain a *controllable* geometric inductive bias that modulates the effective volume growth and representational capacity of the latent space. We hypothesize that this can mitigate mode collapse, and improve sample diversity. We empirically evaluate this hypothesis by sweeping curvature and observing both sample quality and diversity across two image datasets through the FID score.

## 2 METHODOLOGY

We propose a latent generative framework that combines a *hyperbolic autoencoding backbone* (HAE; Li et al. (2023)) with *Riemannian Flow Matching* (RFM; Chen & Lipman (2024)) on the learned hyperbolic latent manifold. HAE uses the $d$-dimensional Poincaré ball model $\mathbb{B}^d_{-1}$, which has negative constant curvature $-1$ and is equipped with Riemannian metric $g_x$, $x \in \mathbb{B}^d_{-1}$. More details about the Poincaré geometry are in Appendix A.2. This model makes use of the exponential and logarithmic maps to convert points from Euclidean to hyperbolic space and vice versa.

**Hyperbolic Autoencoder (HAE).** Following Li et al. (2023), we first invert an image $x_i \in \mathcal{X}$ into the $\mathcal{W}^+$ latent space of a fixed pre-trained StyleGAN2 generator $G$ (Karras et al., 2020) using a fixed pre-trained pSp encoder (Richardson et al., 2021): $\mathbf{w}_i = \mathrm{pSp}(x_i) \in \mathbb{R}^{18 \times 512}$, such that $\mathbf{w}_i \in \mathcal{W}^+$ is the Euclidean latent representation of image $x_i$. Then to obtain a hyperbolic latent code $z_{\mathbb{B}i}$, HAE applies an MLP encoder $\mathrm{MLP}_E$ to reduce dimensionality to $\mathbb{R}^{512}$, followed by a hyperbolic lift using the exponential map at the origin:

$$z_{\mathbb{B}i} = f^{\otimes_c}\big(\exp^c_{\mathbf{0}}\big(\mathrm{MLP}_E(\mathbf{w}_i)\big)\big) \in \mathbb{B}^{512}_c, \tag{1}$$

where $f^{\otimes_c}$ denotes a hyperbolic feed-forward transformation implemented as a Möbius (hyperbolic) linear layer as in Ganea et al. (2018) (see Appendix A.3.1). To reconstruct, the hyperbolic latent is mapped back to a Euclidean tangent space via logarithmic map, then expanded to $\mathcal{W}^+$ with an MLP decoder and then decoded back to pixel space with the STYLEGAN2's generator G:

$$\mathbf{w}'_i = \mathrm{MLP}_D(\log^c_{\mathbf{0}}(z_{\mathbb{B}i})), \qquad x'_i = G(\mathbf{w}'_i). \tag{2}$$

Only the modules between the fixed pSp encoder and the fixed StyleGAN2 generator ($\mathrm{MLP}_E$, the hyperbolic layers, and $\mathrm{MLP}_D$) are trained.

HAE is trained to reconstruct images faithfully and organize the hyperbolic latents *hierarchically*. To achieve this, the loss consists of two reconstruction terms in image space ($\mathcal{L}_2$ for pixel-level reconstruction and $\mathcal{L}_{\mathrm{LPIPS}}$ for perceptual similarity (Richardson et al., 2021)), a reconstruction term in $\mathcal{W}^+$ space ($\mathcal{L}_{\mathrm{rec}}$), and a supervised hyperbolic classification loss ($\mathcal{L}_{\mathrm{hyper}}$), to encourage separability between categories in hyperbolic space, pushing latents of different classes away from each other and latents of the same class together. Importantly, the hyperbolic loss also drives training samples toward the ball boundary, where the space provides greater representational capacity, allowing finer class separation. Detailed loss functions descriptions can be found in Appendix A.3.2.

**Riemannian Flow Matching on the Hyperbolic Latent Manifold.** After training HAE, we obtain a dataset of hyperbolic latents $\mathcal{Z} = \{z_{\mathbb{B}i}\}_{i=1}^N \subset \mathbb{B}^d_c$. We then train a Riemannian Flow Matching model (Chen & Lipman, 2024) to learn a time-dependent tangent vector field $v^t_\theta(\cdot)$ that transports samples drawn from an isotropic wrapped normal distribution on $\mathbb{B}^d_c$ toward the empirical distribution of the training latents, where target samples are defined as $z_1 \sim \mathcal{Z}$.

We define the base distribution $p_0$ as an isotropic wrapped normal centred at the origin of the Poincaré ball (Appendix A.4, Mathieu et al. (2019)). Sampling is performed by drawing a Euclidean normal vector in the tangent space at mean $\mu \in \mathbb{B}^d_c$ and mapping it to the manifold via the exponential map:

$$z_0 = \exp^c_\mu\left(\frac{v}{\lambda_\mu}\right), \qquad v \sim \mathcal{N}(\cdot|0, \Sigma). \tag{3}$$

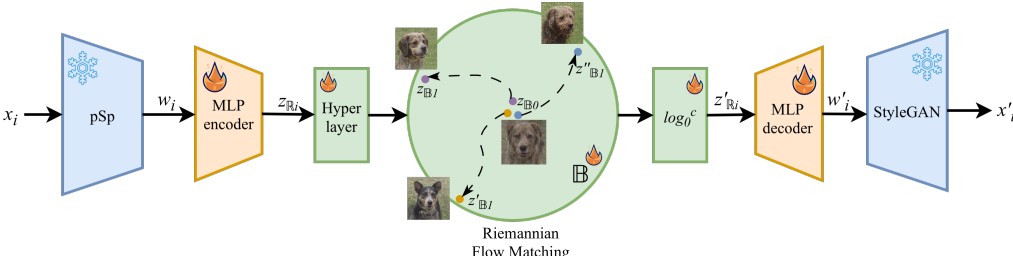

Figure 1: HAE is trained using pretrained pSp encoder and StyleGAN2 decoder. Then the learned Poincaré embeddings are used as samples for the target distribution $(z_{\mathbb{B}1}, z'_{\mathbb{B}1}, z''_{\mathbb{B}1})$, while $z_{\mathbb{B}0}$ are sampled from the wrapped normal at the origin. The learned flow can then be used to generate new hyperbolic latents, which are then mapped back to Euclidean space and decoded by StyleGAN2, to produce new image samples.

For the conditional path construction, we follow Chen & Lipman (2024) and use the geodesic distance as the premetric on the latent manifold. For $t \in [0, 1]$ we can express any interpolation point in closed form via the exponential and logarithmic maps, $z_t = \exp_{z_0}^c \left( t \log_{z_0}^c (z_1) \right)$, which enables simulation-free training of the conditional targets on hyperbolic latents. The RFM loss can then be expressed as

$$\mathcal{L}_{\text{RFM}} = \mathbb{E}_{t, z_0, z_1} \left[ \left\| v_\theta^t(z_t) - \frac{\log_{z_t}^c(z_1)}{1 - t} \right\|_{g_{z_t}}^2 \right]. \tag{4}$$

**Inference.** At inference time, we sample $z_0 \sim p_0$ using the wrapped normal distribution (Eq. 19) centred at the origin, using a small standard deviation to enforce the starting samples near the origin. We then generate a new sample by integrating the learned time-dependent vector field directly on the manifold, ensuring that the trajectories remain in $\mathbb{B}_c^d$:

$$\frac{dz_t}{dt} = v_\theta^t(z_t) \in T_{z_t} \mathbb{B}_c^d, \qquad z_{t=0} = z_0, \qquad z_{t=1} = z_1^*, \tag{5}$$

yielding a latent sample following the learned hyperbolic embedding distribution. The generated latent is then decoded using the HAE decoder and the StyleGAN2 generator to obtain the final image sample. A complete overview of the framework is shown in Figure 1.

## 3 EXPERIMENTS AND RESULTS

We evaluate the proposed framework through controlled synthetic experiments (Appendix A.5), image generation on real datasets, and comparisons against Euclidean latent flow matching. To allow full reproducibility, the implemented code is available at `https://github.com/federicavaleau/Hyperbolic-Flow-Matching`.

We evaluate generation performance on the Animal Faces (Liu et al., 2019) and Flowers (Nilsback & Zisserman, 2008) datasets. We first train HAE using the experimental settings of Li et al. (2023), then we used the training splits of the datasets to extract hyperbolic latent samples for training the Riemannian Flow Matching framework. After training, we sampled from the learned distribution the same amount of samples as the test sets to generate new transported latents, which we then decoded via the HAE decoder, yielding generated images exhibiting fine-grained visual details. Leveraging the publicly available pretrained HAE weights, training the Riemannian Flow Matching component requires only approximately 2.5 hours on a single H100 GPU, demonstrating that generation can be achieved with relatively lightweight additional training.

To assess the benefit of geometry-aware transport, we also train a Euclidean Flow Matching model directly in the StyleGAN2 $\mathcal{W}^+$ latent space, learning a time-dependent Euclidean velocity field transporting samples from an isotropic Gaussian prior, toward the empirical distribution of training $\mathcal{W}^+$ latents. Notably, $\mathcal{W}^+$ has dimensionality $18 \times 512$, whereas the hyperbolic latent space has dimensionality $512$. Generated samples are then decoded with the StyleGAN2 generator. The resulting images exhibit noticeably poorer visual quality and substantially higher FID (Heusel et al.,

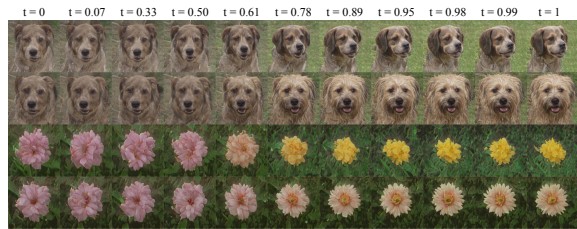 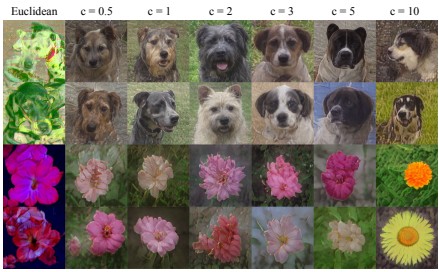

Figure 2: Four decoded samples along the learned paths on the animal faces and flowers datasets. Coarse attributes such as animal species and flowers colors evolve smoothly, illustrating how hyperbolic latent representations organize samples hierarchically.

Figure 3: Generated samples under different curvature values ($c = 0.5, 1, 2, 3, 5, 10$), with Euclidean latent transport shown in the first column.

2017) (Table 1), suggesting that Euclidean transport produces latent codes outside the region of latent space on which the generator was trained, leading to off-distribution decoding artifacts. We therefore hypothesize that as $\mathcal{W}^+$ is not bounded, contrary to the hyperbolic space, and it does not offer an optimal space for efficient latent space learning.

In contrast, the bounded hyperbolic latent space by construction allocates more representational capacity as moving away from the origin, enabling better separation of coarse and fine semantic features, while keeping latent trajectories within regions that decode into more coherent images. Performing flow matching directly on this geometry, allows the generative transport to respect the hierarchical structure previously learned by HAE. In Figure 2, it can be noticed how the hyperbolic flow matching trajectories reflect the hierarchical organization induced by the latent representation, where specific variations emerge along different transport paths.

We observed that generation quality is partly limited by latent capacity. For curvature $c = 1$, the effective radius reachable in the Poincaré ball is set to $\approx 6$ in geodesic distance to avoid numerical instability (Li et al., 2023), restricting how far latent samples can spread while remaining numerically stable. Because curvature defines the latent geometry itself, changing $c$ requires retraining the hyperbolic autoencoder to adapt representations to the new manifold geometry. Increasing curvature changes how volume grows with radius, as in hyperbolic space $\mathbb{H}^n(-c)$, the area enclosed by a circle of radius $r$ is

$$\text{Area}(r) = \frac{2\pi}{c}\left(\cosh(\sqrt{c}\,r) - 1\right), \tag{6}$$

which grows exponentially with curvature. Empirically, we observe that moderate curvature improves realism, while larger curvature increases visual diversity, particularly in backgrounds and color variations. However, excessively large curvature degrades coherence, revealing a trade-off between diversity and transport stability. These results suggest that curvature can act as a controllable geometric inductive bias, contributing to generation diversity. Qualitative results are presented in Figure 3 and Appendix A.5, while quantitative results are reported in Table 1.

Table 1: FID scores for generated samples under different curvature values. Lower is better.

| Curvature ($c$) | 0 | 0.5 | 1 | 2 | 3 | 5 | 10 |
|---|---|---|---|---|---|---|---|
| Animals FID ↓ | 215.62 | 90.62 | **81.89** | 83.05 | 131.29 | 95.09 | 110.35 |
| Flowers FID ↓ | 170.38 | 109.51 | **96.36** | 117.79 | 116.20 | 139.45 | 98.49 |

## 4 CONCLUSIONS AND FUTURE WORK

We presented a geometry-aware generative framework bridging hyperbolic representation learning with Riemannian Flow Matching, showing how the underlying latent geometry influences generative transport. Learning flows directly in hyperbolic latent space preserves hierarchical semantic organization while requiring only lightweight training when combined with a pretrained autoencoder. Qualitative analysis shows that learned trajectories correspond to smooth semantic refinements, reflecting the autoencoder's hierarchical organization. Compared to Euclidean latent transport, hyper-

bolic flows remain within regions that decode into coherent images, yielding more stable samples and motivating further investigation on how other manifold geometries (Davidson et al., 2018) can inform generative modeling. We further investigated curvature as a controllable geometric inductive bias, observing that moderate curvature improves realism while larger curvature introduces more diverse visual details, revealing a trade-off between representational spread and transport stability. Future work may further explore how changes in curvature scale to larger datasets and extend the framework toward class-conditional generation for dataset enrichment in downstream tasks.

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

# A  APPENDIX

## A.1  RELATED WORK

**Encoding Hierarchies.**   Several previous works have explored the benefits of exploiting the intrinsic data's hierarchical structure for various tasks, ranging from different domains such as molecular generation (Qu & Zou, 2024) to image representation (Desai et al., 2023; Liu et al., 2020; Pal et al., 2025) and generation (Li et al., 2023). More specifically, all the previously mentioned works proposed different ways of encoding data in hyperbolic space: Liu et al. (2020) adapted standard CNN pipelines for zero-shot recognition, by taking Euclidean features and mapping them into the Poincaré ball, while replacing Euclidean operations with hyperbolic ones. Desai et al. (2023) and Pal et al. (2025) proposed a combined approach with text and images, respectively MERU and HyCoCLIP. MERU adopts two separate encoders for text and images and then projects their outputs into a shared hyperbolic space, optimizing a contrastive alignment objective. HyCoCLIP additionally introduces a compositional component: together with text and images, localized image regions with associated text phrases are given in input to the encoders, so that the model is able to learn the existing hierarchy across global and local representations in a shared hyperbolic space. These works provide optimal embeddings for image-text retrieval and scene understanding, but they have not been designed for generation.

Qu & Zou (2024) instead, proposes a full hyperbolic autoencoder architecture (HAEGAN) that, exploiting the hierarchical structure of the latent space, is able to generate new molecules and tree-like graphs in latent space with a hyperbolic GAN, and then decode the new samples back to the data domain. Regarding the vision domain, Li et al. (2023) proposed a Hyperbolic Autoencoder (HAE) which is capable of performing 1-shot image generation by perturbing the hyperbolic latents corresponding to training images, resulting in new samples of the desired class. As this autoencoder was designed for visual data, it naturally aligns with the requirements of our setting. We therefore adopt it as the backbone of our approach, on top of which we build our contributions.

**Transport-based Generative Models in Latent Space.**   A growing family of generative models can be unified under the umbrella of transport-based generative models, where sampling is framed as transporting a simple base distribution to a target distribution by learning dynamics defined by an ODE/SDE or an equivalent time-dependent vector field. This field includes normalizing flows (Rezende & Mohamed, 2015) and continuous normalizing flows (Mathieu & Nickel, 2020; Onken et al., 2021), diffusion/score-based models (Ho et al., 2020; Pang et al., 2020; Song et al., 2021), and more recently flow matching (Lipman et al., 2023). To reduce computational cost and better match the intrinsic geometry of the data manifold, Vahdat et al. (2021) proposes to apply score-based generative models in latent space, leveraging a variational autoencoder framework.

Recent works have been exploring the use of a hyperbolic latent representation for generative tasks, using transport-based models. Bose et al. (2020) extend normalizing flows to hyperbolic spaces to construct expressive distributions over hierarchical latent variables. Their contribution involves building coupling transforms operating on the tangent bundle and introduces a wrapped transformation on the hyperboloid model, enabling expressive posteriors with efficient sampling, while respecting hyperbolic geometry.

Another work focused on tree-structured graph generation is Fu et al. (2024), which instead proposes a latent diffusion framework. Instead of diffusing in a Euclidean latent, they construct a hyperbolic latent space with interpretable radial/angular components and design diffusion dynamics constrained along these geometric degrees of freedom to better preserve topological properties during generation. Li et al. (2025), building from the HAE architecture (Li et al., 2023), proposes HypDAE, which combines an hyperbolic autoencoding stage with diffusion-based generation to model hierarchical relationships among seen categories and to enable controllable generation for novel categories. In particular, the hyperbolic latent representation learned by the autoencoder is used as contextual conditioning for the generative process; however, the diffusion dynamics themselves are defined and executed in Euclidean space. As a result, while hierarchical structure is captured at the representation level, the generative transport does not explicitly follow the underlying hyperbolic geometry of the latent manifold.

Finally, Dao et al. (2023) proposes flow matching in the latent space of a pretrained autoencoder. By learning a time-dependent velocity field that transports an isotropic Gaussian prior to the latent codes

of real data, this approach substantially improves computational efficiency and scalability for image generation. This work exploits flow matching specifically in a learned latent embedding derived from an autoencoder, however it still operates in a Euclidean latent space and does not explicitly enforces a hierarchical manifold latent structure, which could improve generation, as showed by other previous transport-based works (Bose et al., 2020; Fu et al., 2024; Li et al., 2025). This gap motivates our hyperbolic flow matching framework, which brings flow matching directly into a hierarchical latent geometry.

## A.2 BACKGROUND IN HYPERBOLIC GEOMETRY

The $d$-dimensional hyperbolic space, $\mathbb{H}^d$, is a homogeneous, simply connected Riemannian manifold with constant negative sectional curvature. Hyperbolic space has the nice property that disc area and circle length grow exponentially with their radius. This enable us to encode infinite layers of semantic hierarchy in a compact way. There are multiple isometric models of Hyperbolic space, each of them presenting its advantages and disadvantages. In this paper, we work with the Poincaré ball model, which is defined as follows. Let

$$\mathbb{B}_c^d = \{x \in \mathbb{R}^d \mid \|x\| < 1\} \tag{7}$$

be the open $d$-dimensional unit ball, where $\|\cdot\|$ denotes the Euclidean norm. The Poincaré ball model of hyperbolic space then corresponds to the Riemannian manifold $(\mathbb{B}_c^d, g_x)$ where

$$g_x = \lambda_x^2 g_E, \quad \lambda_x = \left(\frac{2}{1 - \|x\|^2}\right), \tag{8}$$

and $x \in \mathbb{B}_c^d$, $g_E = \mathbf{I}^d$ denotes the Euclidean metric tensor and $\lambda_x$ denotes the conformal factor. The distance between points $u, v \in \mathbb{B}_c^d$ is given by

$$d_{\mathbb{B}}(u, v) = \operatorname{arcosh}\left(1 + \frac{2\|u - v\|^2}{(1 - \|u\|^2)(1 - \|v\|^2)}\right). \tag{9}$$

This model makes use of the exponential $\exp_x^c(\cdot) : T_x\mathbb{B}_c^d \to \mathbb{B}_c^d$ and logarithmic $\log_x^c(\cdot) : \mathbb{B}_c^d \to T_x\mathbb{B}_c^d$ maps to convert points from Euclidean to hyperbolic space and vice versa, where $T_x\mathbb{B}_c^d$ denotes the tangent space of $\mathbb{B}_c^d$ at $x$, as the first order linear approximation of $\mathbb{B}_c^d$ around $x$. More explicitly, for $v \in T_x\mathbb{B}_c^d$, $y \in \mathbb{B}_c^d$ and $\oplus_c$ denoting the Möbius addition defined in Khrulkov et al. (2020)

$$\exp_x^c(v) := x \oplus_c \left(\tanh\left(\sqrt{c}\frac{\lambda_x\|v\|}{2}\right)\frac{v}{\sqrt{c}\|v\|}\right), \tag{10}$$

$$\log_x^c(y) := \frac{2}{\sqrt{c}\lambda_x} \operatorname{arctanh}(\sqrt{c}\| - x \oplus_c y\|)\frac{-x \oplus_c y}{\| - x \oplus_c y\|} \tag{11}$$

The geodesics in this manifold (these are, the paths of locally shortest length connecting two points) appear as arcs of circles that are orthogonal to the boundary of the ball or as straight lines passing through the centre. This model provides a nice interpretability of the latent space used for generation, as shown in Figure 4.

## A.3 HYPERBOLIC NETWORKS

### A.3.1 HYPERBOLIC FEED-FORWARD LAYER

To define hyperbolic feed-forward layers on the Poincaré ball $\mathbb{B}_c^d$, Ganea et al. (2018) propose to lift standard Euclidean maps to the manifold by composing them with the logarithm and exponential maps at the origin. This yields an analogue version of Euclidean linear layers and enables the application of pointwise nonlinearities in hyperbolic space.

For a Euclidean map $f : \mathbb{R}^d \to \mathbb{R}^m$, its Möbius version is the map $f^{\otimes_c} : \mathbb{B}_c^d \to \mathbb{B}_c^m$ defined by

$$f^{\otimes_c}(x) := \exp_0^c\big(f(\log_0^c(x))\big), \tag{12}$$

where $\log_0^c$ and $\exp_0^c$ are the logarithm and exponential maps at the origin. The Euclidean mapping can be recovered for curvature approaching 0 and continuous $f$, i.e., $\lim_{c \to 0} f^{\otimes_c}(x) = f(x)$.

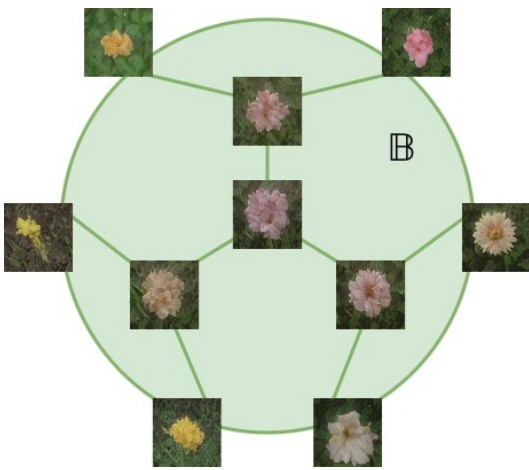

Figure 4: Illustration of the learned Poincaré latent space organization. Samples near the origin correspond to coarse, highly generic flower representations, while moving radially outward yields increasingly specific and visually detailed instances. Points close to the boundary represent fine-grained samples, including training images, reflecting how hyperbolic geometry naturally organizes representations from general concepts at the center to higher granularity toward the boundary.

### A.3.2 HAE LOSS FUNCTIONS

For the pixel-level reconstruction, we use the pSp losses, where the $\mathcal{L}_{\text{LPIPS}}$ loss is a perceptual loss computed with a fixed feature extractor $F(\cdot)$ (Richardson et al., 2021):

$$\mathcal{L}_2(x_i) = \|x_i - x_i'\|_2, \tag{13}$$

$$\mathcal{L}_{\text{LPIPS}}(x_i) = \|F(x_i) - F(x_i')\|_2. \tag{14}$$

To ensure the hyperbolic bottleneck can be decoded back to the original StyleGAN2 latent, we ensure that $\mathbf{w}_i'$ matches to $\mathbf{w}_i$:

$$\mathcal{L}_{\text{rec}}(\mathbf{w}_i) = \|\mathbf{w}_i - \mathbf{w}_i'\|_2. \tag{15}$$

To enforce that hyperbolic codes respect semantic hierarchy, HAE employs a multinomial logistic regression defined directly on the Poincaré ball (the hyperbolic softmax from Ganea et al. (2018)). Each class $k$ is associated with a reference point $p_k \in \mathbb{B}_c^d$ and a tangent vector $a_k \in T_{p_k}\mathbb{B}_c^d$ defining a hyperbolic hyperplane. Classification probabilities are obtained by measuring the signed distance of a latent point to these hyperplanes in hyperbolic space. For a latent code $x \in \mathbb{B}_c^d$, the class probability is given by

$$p(y = k \mid x) \propto \exp\left( \text{sign}(\langle -p_k \oplus_c x, a_k \rangle) \sqrt{g_{p_k}(a_k, a_k)}\, d_{\mathbb{B}}(x, \tilde{H}_{a_k, p_k}^c) \right), \tag{16}$$

where $\tilde{H}_{a_k, p_k}^c$ denotes the hyperbolic hyperplane defined by $(p_k, a_k)$ (Ganea et al., 2018). Training minimizes the negative log-likelihood

$$\mathcal{L}_{\text{hyper}} = -\frac{1}{N} \sum_{n=1}^{N} \log p(y_n \mid x_n), \tag{17}$$

where $y_n$ denotes the ground-truth class of sample $x_n$.

Then the overall loss function, including adaptive parameters $\lambda_1$, $\lambda_2$ and $\lambda_3$ is:

$$\mathcal{L}_{\text{HAE}} = \mathcal{L}_2 + \lambda_1 \mathcal{L}_{\text{LPIPS}} + \lambda_2 \mathcal{L}_{\text{rec}} + \lambda_3 \mathcal{L}_{\text{hyper}}. \tag{18}$$

### A.4 WRAPPED NORMAL DISTRIBUTION

A wrapped normal density function centered at $\mu$ can be defined as:

$$p_0(z) = \mathcal{N}_{\mathbb{B}^d}^w(z \mid \mu, \sigma^2) = (2\pi\sigma^2)^{-d/2} \exp\left( -\frac{d_{\mathbb{B}}(\mu, z)^2}{2\sigma^2} \right) \left( \frac{d_{\mathbb{B}}(\mu, z)}{\sinh(d_{\mathbb{B}}(\mu, z))} \right)^{d-1}, \tag{19}$$

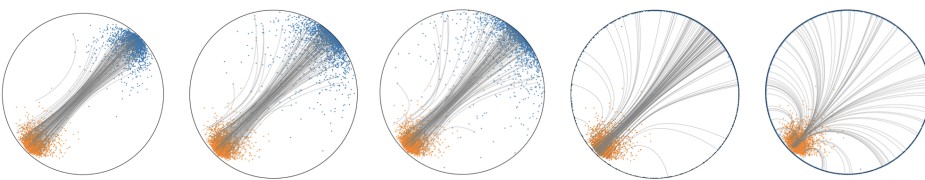

Figure 5: Learning paths from a starting wrapped normal distribution with increasingly higher standard deviations (blue points) to a goal wrapped normal distribution with fixed standard deviation (orange points). Gray lines show trajectories induced by the learned flow, which approximate geodesic transport between source and target point.

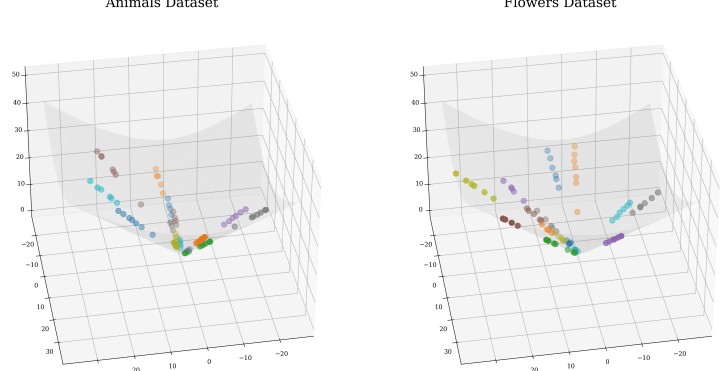

Figure 6: 3D projections on the hyperboloid of the sampled latents shown in Figures 7 and 8. Each color corresponds to a distinct trajectory transporting samples from the base distribution toward different regions of the learned data distribution. The structured paths illustrate how transport dynamics organize samples along coherent semantic directions in the hyperbolic latent space, starting near the origin and flowing towards regions encoding increasingly fine-grained representations.

and its log-density is then given by:

$$\log p_0(z) = -\frac{d}{2}\log(2\pi\sigma^2) - \frac{d_{\mathbb{B}}(\mu, z)^2}{2\sigma^2} + (d-1)\Big(\log d_{\mathbb{B}}(\mu, z) - \log\sinh(d_{\mathbb{B}}(\mu, z))\Big), \quad (20)$$

which is used for the loglikelihood computation during flow matching validation. Exact likelihoods are computed by integrating the probability flow ODE backward from a latent data point $z_1$ to its corresponding base point $z_0$. Along the trajectory, the log-density changes according to the accumulated divergence of the flow

$$\log p_1(z_1) = \log p_0(z_0) - \int_0^1 \mathrm{div}_g\big(v_\theta^s\big)(z_s)\, ds, \quad (21)$$

where $\mathrm{div}_g$ denotes the Riemannian divergence induced by the manifold metric.

## A.5 ADDITIONAL RESULTS

**Toy Data.** Before generalizing the wrapped normal distribution to higher dimensions, we first evaluated the transport mechanism on a synthetic 2D example, learning a flow between two distributions located on opposite sides of the Poincaré disk and with progressively different standard deviations. The learned Riemannian flow accurately matches the target distribution, confirming that the model correctly captures manifold transport dynamics (Figure 5).

**Image Data.** Figures 7 and 8 show additional samples along the learned Riemannian flow in hyperbolic space. A projection of the sampled trajectories in 3D, performed using UMAP (McInnes et al., 2018), is shown in Figure 6. Figures 9 and 10 show randomly sampled images from the learned distributions under different curvatures ($c = 0.5, 1, 2, 3, 5, 10$) and in the Euclidean setting.

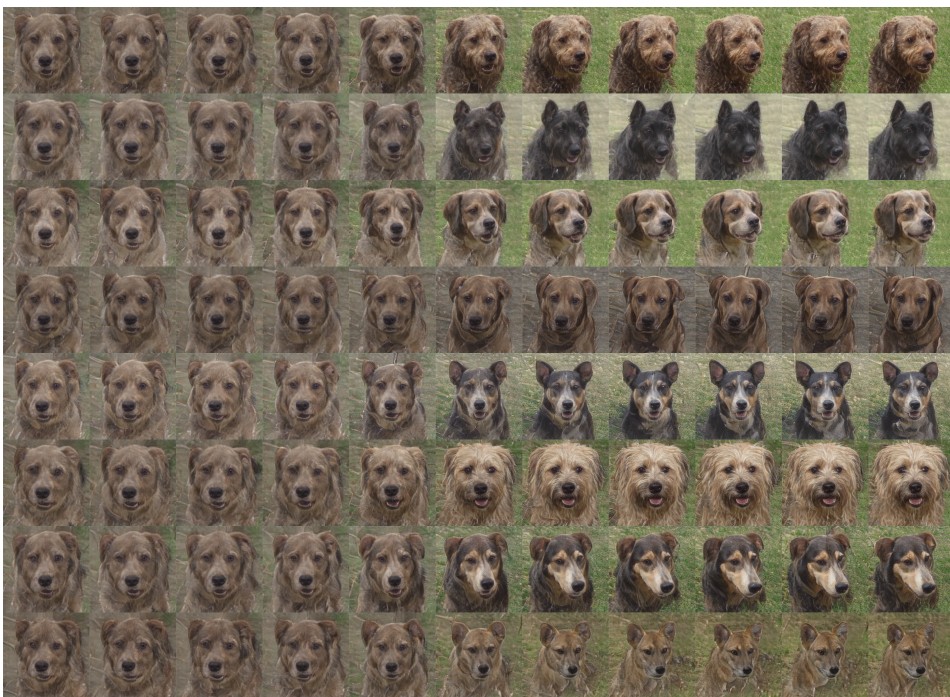

Figure 7: Eight decoded samples along the learned paths on the animal faces dataset, sampling started from $z_{t=0} \sim \mathcal{N}_{\mathbb{B}^d}^w(\mathbf{0}, 0.03)$ to $z_{t=1}$. Coarse attributes such as animal species and colors evolve smoothly while finer details vary progressively, illustrating how the hyperbolic latent representation organizes samples hierarchically, with shared global characteristics preserved near the origin and more specific variations emerging along the flow.

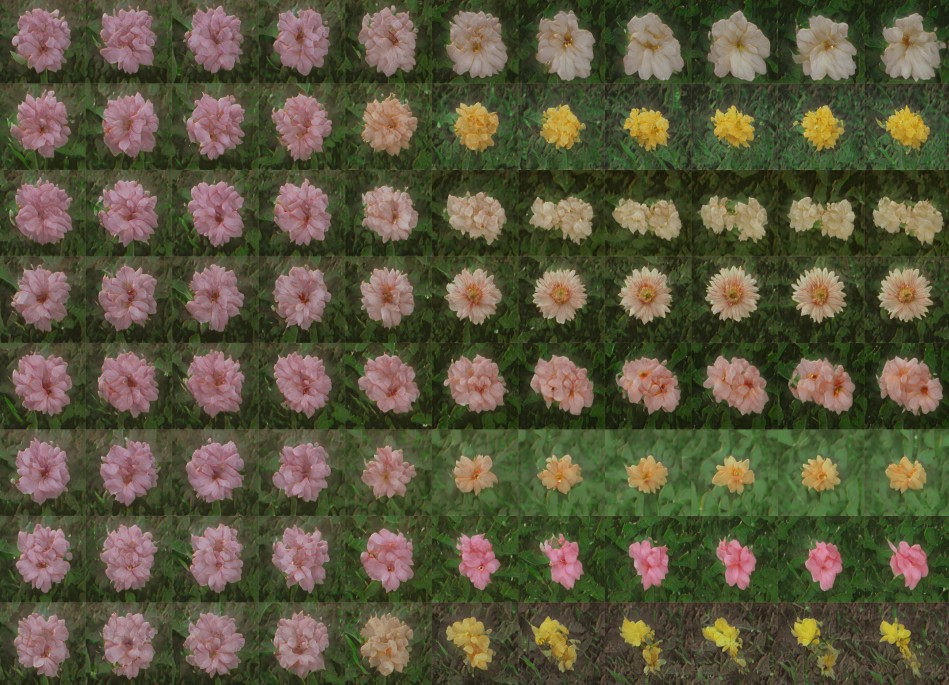

Figure 8: Eight decoded samples along the learned paths on the flowers dataset, sampling started from $z_{t=0} \sim \mathcal{N}_{\mathbb{B}^d}^w(\mathbf{0}, 0.03)$ and ended in $z_{t=1}$. Coarse attributes such as global flower structure and color evolve progressively during the flow, revealing the structure of the navigated hyperbolic space.

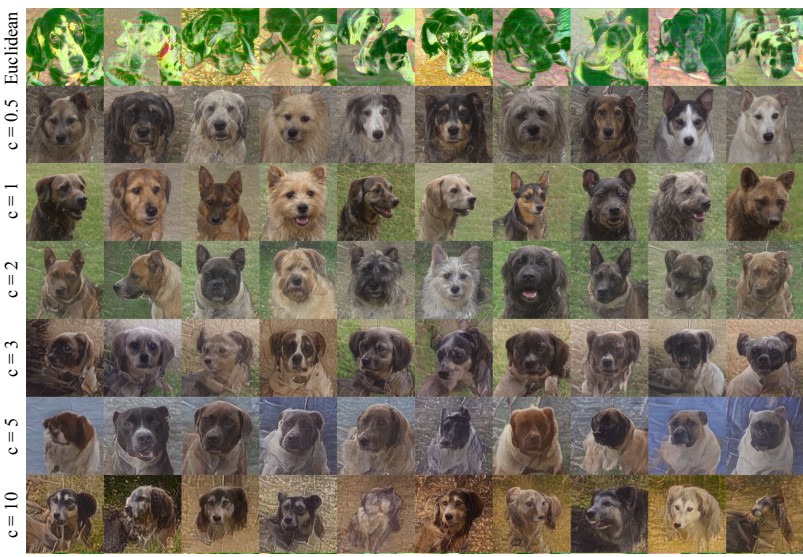

Figure 9: Moderate curvature values yield the best balance between realism and diversity: facial structure remains coherent while variations in pose and fur texture increase across samples. Interestingly, larger curvature values (i.e., $c = 3, 5$) introduce greater variability in background and color range, but reduce structural coherence. In contrast, Euclidean latent transport produces visibly distorted or off-manifold samples, indicating unstable latent trajectories.

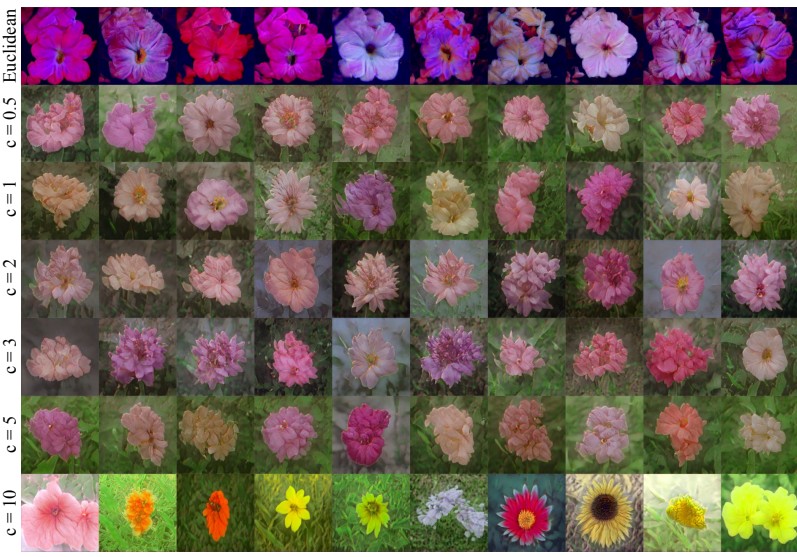

Figure 10: Multiple curvatures produce visually coherent flowers, smaller curvatures exhibit less variation in background colors while allowing meaningful but moderate variation in petal shape and color. As curvature increases (i.e., $c = 2, 3, 10$), diversity grows in background composition and with $c = 10$, substantially different flower shapes appear. Euclidean transport instead produces samples with homogeneous structure and artifacts.

