# OpenReview forum: "Hyperbolic Curvature as an Inductive Bias for Latent Space Flow Matching"
_ICLR.cc/2026/Workshop/GRaM — ICLR 2026 Workshop GRaM Poster_

### Official Review · Reviewer_UEkn · 2026-02-23
**Promising Geometric Framework Undermined by Absent Baselines and Week Quantitative Evaluation**

**Rating:** 3
**Confidence:** 3

**Review:**

**Strengths:**

- Strong motivation in combining hyperbolic autoencoder and Reimannian Flow matching into a geometry-aware framework with clear intuitive justification for why a hyperbolic latent space is preferable to Euclidean for hierarchical image data.
- The curvature as an inductive bias framing is novel and an interesting contribution and the empirical sweep across curvature values provides evidence of a diversity realism tradeoff.
- Training cost was relatively light weight with only 2.5 hours on a single H100 building on pretrained HAE and making it more practicable and having a lower barrier for reproduction.

**Weakness:**
- No comparison to standard generative model baselines (e.g., StyleGAN2, StyleGAN3, diffusion models) is provided making it impossible to contextualize whether the proposed framework achieves competitive generation quality. The Euclidean flow matching baseline is described qualitatively as achieving substantially higher FID, but the actual scores are either absent or ambiguously reported in Table 1, making the primary comparison unverifiable.
- The best reported FIDs of 81.89 and 96.36 on the Animals and Flowers dataset are poor in absolute terms for image generation, and without external baselines there is no way to determine where these reflect meaningful results.
- Despite claiming diversity as a claim there is no quantitative diversity metric used (e.g. recall, coverage) are reported, with FID allowing conflating quality and diversity and being insufficient to support this claim.

**Questions:**
- What FID scores do standard generative baselines such as StyleGAN2 or a diffusion model achieved on the Animals and Flowers dataset, and how does the proposed framework compare?
- How would the framework perform on more general image generation datasets such as ImageNet or CIFAR, rather than on fine-grained and visually homogenous datasets?

**Pmlr Suitability:**

NA

---

### Official Review · Reviewer_HKnf · 2026-02-23
**Technically correct method with weak evaluation**

**Rating:** 5
**Confidence:** 5

**Review:**

# Summary:

This paper proposes an approach for generative modeling using latent flow matching in a hyperbolic space.  The problem the authors tackle is that natural image data has hierarchical structure that is not adequately captured generative models with a Euclidean latent space.  To solve this problem, the paper proposes to instead construct a generative model that has a hyperbolic latent space because hyperbolic spaces can naturally encode hierarchical structures, and because this approach has had success in many other applications including other generative modeling techniques.  The authors are the first to do this in the specific setting of latent flow matching.  The authors develop this approach and show that the curvature of the latent space is correlated with the generative performance of the model.

# Strengths:
- The approach is technically sound.
- This specific approach is novel.
- The paper is well written and easy to follow.
- The results demonstrate that there is an interesting relationship between the curvature of the latent space and the visual quality of the generated samples, as measured by FID score.

# Weaknesses:
- The experimental evaluation against the Euclidean space baseline seems wrong.  It does not seem correct to compare the hyperbolic model, which performs flow matching in a 512 dimensional latent space (as per line 071) against a model where flow matching is done in $\mathcal{W}^+$ space, which is 18x512 dimensional.

- The qualitative experiments are not convincing.  For example, I'm not sure that I agree that Figure 2 demonstrates that the coarse attributes evolve more smoothly than what I would expect a regular Euclidean latent flow matching trajectory to look like.  Additionally, I'm not even sure that this kind of experiment is correct for demonstrating that the latent hyperbolic representation organize samples hierarchically as it has more to do with the generation procedure itself rather than the geometry of the model.

- The approach is not too novel as building generative models with hyperbolic latent spaces has been done before in a few different contexts (as noted by the authors in the related work), and the technical approach to do this for latent flow matching has been known in the community through the hyperbolic autoencoder (Li et al. (2023)) and flow matching on general geometries (Chen & Lipman (2024)).

Minor:
- The relationship between the curvature parameter, c, is never explicitly related to the exponential and logarithmic maps in appendix A2.

**Pmlr Suitability:**

NA

---

### Meta-Review · Area_Chair_vTzy · 2026-02-27

**Decision:**

Reject

**Metareview:**

The paper provides an approach for generative modeling where latent flow matching is done in a hyperbolic space. The arguments for the desirability of operating in such a space are standard and have appeared in the ML literature from time to time. The approach developed in the paper is interesting. However, the reviewers concur that slightly better experimental validation would have pushed the paper over the mark. I think the paper has value to be presented at the workshop, but from the write-up the benefit afforded by the framework is not entirely clear.

**Relevance To Proceedings:**

Tiny paper — does not apply

**Relevance To Workshop:**

Yes — suitable for GRaM

---

### Decision · Program_Chairs · 2026-03-02

**Decision:**

Accept (Poster)

**Comment:**

We appreciate the reviewers comments, although the goal of Tiny paper at this workshop is not to show state-of-the-art results necessarily, but to use geometry for meaningful insights. We agree that further experiments would strengthen the work, but given the novelty of the approach, it’s enough to accept the paper.